# Tracing the evolution of key traits in dorid nudibranchs

**Silvia Prieto-Baños[1,2,3], Kara K. S. Layton**[1,4]*

1 School of Biological Sciences, University of Aberdeen, Aberdeen, United Kingdom, 2 Department of Computational Biology, University of Lausanne, Lausanne, Switzerland, 3 SIB Swiss Institute of Bioinformatics, Lausanne, Switzerland, 4 Department of Biology, University of Toronto Mississauga, Mississauga, Ontario, Canada

* kara.layton@utoronto.ca

## Abstract

Reconstructing trait evolution is critically important for elucidating the processes generating biodiversity. However, this work is in its infancy in non-model clades for which we lack a basic understanding of their ecology and biology. Here, we compile information about prey preference, chemical acquisition and colour pattern in dorid nudibranchs (Nudibranchia: Doridoidei) and reconstruct their ancestral states using a multi-gene phylogeny to investigate the evolution of these key traits. Our analyses show that the most recent common ancestor (MRCA) of Doridoidei preferred sponge prey from which they sequestered metabolites, and subsequent shifts to different prey types and *de novo* synthesis of defensive compounds occurred multiple times independently across the phylogeny. Additionally, the MRCA likely exhibited complex colour patterns, including spots or stripes, with uniform morphotypes evolving in most families. Despite the fact that many dorid nudibranchs derive both metabolites and pigments from their prey, we found no evidence of correlated evolution amongst these traits. As part of this work, we present a multi-gene phylogeny for Doridoidei with representatives from 88 genera and 18 families, but there remain issues with poor support across the tree. Nonetheless, for the first time, we explore the evolution of key traits that contributed to the diversification of dorid nudibranchs, highlighting the need for more refined trait data and greater phylogenetic resolution for future work.

## 1. Introduction

Understanding the ecological factors underpinning speciation and the generation of biodiversity is of utmost importance in evolutionary biology. Despite this, we lack the fundamental ecological knowledge required for this work, and this is especially true for understudied invertebrate groups. Nudibranch molluscs (Gastropoda: Heterobranchia) exhibit remarkable diversity on multiple scales and serve as unique and interesting systems in ecology and evolutionary biology. The Doridoidei infraorder (herein referred to as dorids) is one of the most speciose nudibranch groups, comprising over 2,000 species [1–3]. Dorids are globally distributed, occupying a wide range of marine environments and ecological niches, and they exhibit diverse behavioural and morphological traits, including dietary specialization [4,5], aposematic colour patterns [6,7] and chemical defense [8,9]. Nudibranchs feed largely on

**Data availability statement:** Trait data is available within the manuscript and the sequence

alignment employed for analysis is available on Dryad: 10.5061/dryad.8sf7m0d0m.

**Funding:** The author(s) received no specific funding for this work.

**Competing interests:** The authors declare that they have no conflicts of interest.

other invertebrates [3] but their feeding preferences, and degree of specialization, likely vary across taxonomic groups and ecological contexts. For instance, Bathydoridoidei, the comparably species poor sister group to Doridoidei, are generalists that feed on multiple marine invertebrate phyla (e.g., echinoderms, bryozoans, crustaceans) [10], while most members of the Doridoidei are thought to have more specialized diets [11]. Broadly speaking, dietary specialization and prey shifts have been invoked as drivers of diversification in marine heterobranchs [5,12], but how chemical defense and colour pattern variation might interact and play a role is largely unknown.

Nudibranchs are known to house diverse chemical compounds that play a key role in predator avoidance [11]. In fact, the bioactive metabolites isolated from nudibranchs have received considerable attention from natural products researchers for their potential use in antiviral and anticancer pharmaceuticals (e.g., [13]). The diverse chemical compounds found in nudibranchs are typically sequestered from their prey or *de novo* synthesized. In the case of sequestration, bioactive compounds can be selectively stored while others are discarded [14], and in some cases inactive compounds may be secondarily modified [13]. For instance, *Chromodoris* nudibranchs selectively sequester bioactive latrunculin A in specialized mantle glands [14], while *Felimida* secondarily modifies sesquiterpene from their sponge prey [15]. Conversely, *Dendrodoris grandiflora* has been shown to *de novo* synthesize the sesquiterpene polygodial [16]. Previous work has characterized chemical diversity and the mode of chemical acquisition in some nudibranchs (e.g., [13,14,17]) but whether this is phylogenetically conserved remains unknown.

Chemically-defended species often exhibit bright colours to advertise their toxicity or unpalatability to predators in a strategy known as aposematism [18]. Nudibranchs display a fascinating diversity of colour patterns and serve as ideal models for understanding aposematism in marine systems. Recent work has shown considerable variation in both chemical defence and warning signals (colour patterns) amongst aposematic nudibranch species, especially those involved in Müllerian mimicry rings- where multiple, toxic species share similar colour patterns (e.g., [9]). Several other studies have shown that colour patterns are unreliable as diagnostic morphological characters since distantly-related species can share similar colour patterns and conversely colour patterns can vary drastically within species [7,19,20]. Despite these studies, our understanding of the evolution of colour patterns in nudibranchs is truly in its infancy, especially compared to terrestrial species (e.g., *Heliconius*, [21]). Across diverse terrestrial taxa, aposematic species tend to display high-contrast spots or bands while cryptic (camouflaged) species tend to exhibit irregular blotches or stippling, and stripes can be used in both aposematism or crypsis depending on environmental complexity [22–24]. These same colour pattern elements appear across diverse nudibranch taxa but whether they serve the same function in marine species is unknown. Prey preference, chemical acquisition and colour pattern have been described and studied across several nudibranch groups (see above), but no study has investigated the evolution of these traits in tandem, especially in a phylogenetic context.

Key to understanding trait evolution is the presence of a robust phylogeny for which to track trait gain and loss over evolutionary time. Here, we reconstruct Doridoidei phylogeny using publicly available sequence data and we employ ancestral state reconstruction to understand how prey preference, chemical acquisition and colour pattern have evolved over time in dorids. In doing so, we discuss major phylogenetic relationships within Doridoidei, we identify the most likely character states of each ancestral node in the tree, and we look for evidence of correlated evolution amongst traits. For the first time, this study investigates the role that ecological traits have played in shaping dorid nudibranch biodiversity and evolution and in doing so, compiles one of the most extensive ecological datasets for dorid nudibranchs to date.

## 2. Methods

### Compiling genetic and trait data

Sequence data and trait data were compiled for 88 genera and 18 families of dorid nudibranch from publicly available resources (Table 1; Fig 1). One representative species per genus was used for all analyses since, in most cases, trait data was only available at this taxonomic rank. The representative taxa chosen for this study were those that had sufficient genetic data to support phylogenetic reconstruction (e.g., data for at least two of five genes where possible). Five molecular markers were employed for phylogenetic inference and ancestral state reconstruction (ASR), including two mitochondrial (COI, 16S) and three nuclear genes (28S, 18S, H3). All molecular data was mined from GenBank for this work and corresponding accession numbers are available in Table 1. Trait data was obtained from research articles and citizen science databases provided in S1 File. Trait data is resolved at the genus-level because it was lacking for many species, but where it was also lacking at the genus-level we either marked this as missing data or we assigned traits based on confamilial data where there was little variation at the family-level (see Table 1). For each dorid genus, its preferred prey was categorized at the phylum-level, except for one genus where prey preference was characterized as 'generalist' because records indicated multiple possible prey sources. Chemical acquisition was categorized as sequestration or *de novo* synthesis, but this information was missing for several taxa. Where secondary modification of a chemical obtained from prey was reported in the literature, we considered these as sequestration. Lastly, colour patterns were categorized as uniform, with spots or stripes, mottled or with a distinct mantle rim. The latter was included here because the distinct, coloured mantle rim (e.g., *Chromodoris*, Fig 1I) may be an important anti-predator visual signal in nudibranchs [6]. Conversely, the mottled colour pattern refers to large, undefined blotches or stippling on the mantle (e.g., *Aphelodoris*, Fig 1R) compared to defined spots (e.g., *Dendrodoris*, Fig 1B). Where tubercles or papilla on the mantle had distinctly coloured tips (e.g., *Cadlinella*, Fig 1F), these were considered visually similar to spots and thus the colour pattern was defined as such. All colour pattern data was retrieved from specimen images on the Sea Slug Forum and iNaturalist. All taxonomic names were verified on the World Register of Marine Species (WoRMS).

### Phylogenetic and ancestral state reconstruction

Sequence data for all five markers was downloaded from GenBank and aligned with MAFFT v7.52 [25] before concatenation in Geneious v.9 (https://www.geneious.com). A maximum-likelihood tree was obtained for this concatenated dataset using IQ-TREEv2.2.0 [26] with a GTR + G + I model and 1000 ultrafast bootstrap replicates and with *Bathydoris* as an outgroup [1]. A Bayesian tree was obtained with MrBayes v3.2 [27] using a GTR + G + I model, 10,000,000 MCMC generations, a 25% burnin and four Markov chains, with trees sampled every 100 generations. As described above, we conducted an extensive literature search to generate trait data for contemporary taxa as input for ASR. The final ML tree was used alongside this trait data for ASR in the *corHMM* package in R [28]. This package, and the rayDISC function specifically, was chosen for ASR since it accepts both missing data and polymorphic character states (e.g., members of some genera can both synthesize and sequester defensive compounds; members of some genera exhibit spots/stripes while others are uniform). This functionality was critically important since nudibranch traits are complex and, in some cases, unknown. We conducted ASR with both Equal Rates (ER) and All-Rates-Different (ARD) models and employed the model that had the lowest corrected Akaike Information Criterion (AICc) score. In all cases, the ER model was the best fit and was executed with default parameters and marginal likelihoods in *corHMM*. Lastly, we tested for correlated evolution among

**Table 1. Sequence data and trait data for all dorid taxa in this study.** GenBank accession numbers are provided for each gene. Prey type is characterized at the phylum-level. Chemical acquisition is characterized as sequestration or *de novo* synthesis. Colour pattern is characterized as uniform, with stripes or spots, mottled or with a distinct mantle rim. References are available in S1 File. Missing data is represented by a dash. Data marked with an asterisk indicates that genus-level information was lacking and trait data was instead assigned based on family-level patterns or supporting data, provided in parentheses.

| Species | GenBank Accessions | | | | | Prey Group | Chemical Acquisition | Colour Pattern |
|---|---|---|---|---|---|---|---|---|
| | COI | 16S | 28S | 18S | H3 | | | |
| **Actinocyclidae** | | | | | | | | |
| *Actinocyclus verrucosus* | MF958438 | MF958311 | MF958397 | MF958352 | – | Porifera [1] | – | Mottled |
| *Hallaxa translucens* | EU982760 | EU982814 | KT698821 | MF958341 | – | Porifera & Chordata[2-4] | – | Uniform |
| **Aegiridae** | | | | | | | | |
| *Aegires flores* | MF958442 | MF958316 | MF958402 | – | – | Porifera[2,3,5-8] | Sequestration [9] | Stripes/spots & distinct rim |
| *Notodoris minor* | KP871631 | KP871678 | – | – | KP871654 | Porifera [10,11] | Sequestration [12,13] | Stripes/spots |
| **Akiodorididae** | | | | | | | | |
| *Akiodoris lutescens* | – | MN224076 | MN224112 | – | – | Bryozoa [14] | – | Uniform |
| *Armodoris anudeorum* | KP340387 | KP340290 | KP340355 | – | KP340412 | – | – | Uniform |
| **Cadlinellidae** | | | | | | | | |
| *Cadlinella ornatissima* | MF958415 | MF958284 | MF958371 | MF958325 | – | Porifera [2] | Sequestration* [15] (presence of MDFs) | Stripes/spots |
| **Cadlinidae** | | | | | | | | |
| *Aldisa sanguinea* | MF958435 | MF958309 | MF958394 | MF958350 | – | Porifera [10] | Sequestration [10] | Uniform |
| *Cadlina laevis* | MN224049 | MN224081 | MN224116 | – | – | Porifera [8, 10, 16-18] | Sequestration [10, 19] & synthesis [10] | Stripes/spots & distinct rim |
| *Inuda luarna* | EU982718 | EU982768 | – | – | – | Porifera* (confamilial trait) | Sequestration* [20] (presence of MDFs) | Uniform |
| **Calycidorididae** | | | | | | | | |
| *Calycidoris guentheri* | KP340397 | KP340301 | KP340371 | KP340338 | KP340417 | Bryozoa* (confamilial trait) | Synthesis* (absence of mantle glands; confamilial trait) | – |
| *Diaphorodoris lirulatocauda* | KP340403 | KP340307 | KP340377 | KP340344 | KP340422 | Bryozoa [2, 8, 21] | Synthesis [22] | Stripes/spots & distinct rim |
| **Chromodorididae** | | | | | | | | |
| *Ardeadoris scottjohnsoni* | EU982714 | EU982763 | KT698766 | – | – | Porifera [23] | Sequestration [24] | Stripes/spots & distinct rim |
| *Ceratosoma brevicaudatum* | – | EU512141 | EU512052 | – | – | Porifera [25, 26] | Sequestration [25, 26] | Stripes/spots & distinct rim |
| *Chromodoris magnifica* | EF535110 | EF534042 | EF534028 | – | – | Porifera [27, 28] | Sequestration [28] | Stripes/spots & distinct rim |
| *Chromolaichma edmundsi* | HM162686 | HM162595 | MF958390 | MF958347 | HM162595 | Porifera [29] | Sequestration [30] | Distinct rim |
| *Diversidoris aurantinodulosa* | EF535141 | EF534069 | – | EF534011 | – | Porifera [5] | Sequestration* (confamilial trait) | Stripes/spots & distinct rim |
| *Doriprismatica stellata* | KT600693 | KT595622 | KT698782 | – | – | Porifera [31] | Sequestration [31] | Mottled & distinct rim |
| *Felimare tema* | HM162685 | HM162594 | MF958389 | MF958346 | HM162594 | Porifera [32] | Sequestration [10, 33] | Stripes/spots & distinct rim |
| *Glossodoris buko* | KT600711 | KT595638 | KT698808 | – | – | Porifera [25] | Sequestration [25, 29] | Stripes/spots, mottled & distinct rim |
| *Goniobranchus reticulatus* | JQ727853 | JQ727733 | – | – | – | Porifera [24, 25] | Sequestration [24, 25] | Stripes/spots, mottled & distinct rim |

*(Continued)*

**Table 1.** (Continued)

| Species | GenBank Accessions | | | | | Prey Group | Chemical Acquisition | Colour Pattern |
|---|---|---|---|---|---|---|---|---|
| | COI | 16S | 28S | 18S | H3 | | | |
| *Hypselodoris obscura* | MG645598 | MG645438 | – | – | MG645519 | Porifera [10] | Sequestration [10] | Stripes/spots, mottled & distinct rim |
| *Mexichromis antonii* | EU982748 | EU982800 | – | – | MG645548 | Porifera [2] | Sequestration* (confamilial trait) | Stripes/spots & distinct rim |
| *Miamira striata* | MW892629 | | – | – | MW883960 | Porifera [10] | Sequestration [10] | Stripes/spots, mottled & distinct rim |
| *Thorunna florens* | JQ727913 | JQ727817 | – | – | – | Porifera [3] | Sequestration* (confamilial trait) | Stripes/spots & distinct rim |
| *Tyrinna evelinae* | EU982757 | EU982811 | MF958391 | – | – | Porifera [18, 34] | Sequestration [10] | Stripes/spots |
| *Verconia verconis* | EF535118 | EF534046 | – | EF534036 | – | Porifera [2] | Sequestration* (confamilial trait) | Stripes/spots, mottled & distinct rim |
| **Corambidae** | | | | | | | | |
| *Corambe obscura* | KP340399 | KP340303 | KP340373 | KP340340 | KP340419 | Bryozoa [2, 35] | – | Uniform |
| **Dendrodorididae** | | | | | | | | |
| *Dendrodoris densoni* | – | MF958308 | MF958393 | MF958349 | – | Porifera [2, 36] | Synthesis [25, 36, 37] | Stripes/spots & mottled |
| *Doriopsilla janaina* | – | MF958312 | MF958398 | MF958353 | – | Porifera [2, 38-40] | Synthesis [10, 40] | Stripes/spots |
| **Discodorididae** | | | | | | | | |
| *Asteronotus cespitosus* | MF958419 | MF958288 | MF958375 | MF958328 | MN720324 | Porifera [9, 41] | Sequestration [9, 41] | Uniform |
| *Atagema kimberlyae* | OQ362152 | – | – | – | OQ366206 | Porifera [2] | – | Stripes/spots |
| *Carminodoris flammea* | MN720285 | – | – | – | MN720311 | – | – | Mottled |
| *Diaulula sandiegensis* | KP871647 | KP871695 | – | – | – | Porifera [2, 10] | Sequestration [10, 42] | Stripes/spots |
| *Discodoris coerulescens* | MF958421 | MF958290 | MF958377 | MF958330 | – | Porifera [2] | Sequestration [43] | Mottled |
| *Geitodoris heathi* | KP871642 | KP871690 | – | – | KP871666 | Porifera [2] | – | Mottled |
| *Halgerda dalanghita* | MF958420 | MF958289 | MF958376 | MF958329 | MN720316 | Porifera [9] | Sequestration [9, 25] | Stripes/spots, mottled & distinct rim |
| *Hoplodoris nodulosa* | FJ917486 | FJ917428 | FJ917469 | FJ917443 | – | – | – | Mottled |
| *Jorunna tomentosa* | AJ223267 | AJ225191 | – | – | – | Porifera [44] | Sequestration [25, 45] | Stripes/spots |
| *Paradoris liturata* | KP871648 | KP871696 | – | – | – | Porifera [43] | – | Stripes/spots |
| *Peltodoris nobilis* | HM162684 | HM162593 | – | – | HM162499 | Porifera [10, 46-48] | Sequestration [10, 48] | Mottled |
| *Platydoris sanguinea* | MF958416 | MF958285 | MF958372 | MF958326 | MN720312 | Porifera [49] | Synthesis [40] | Mottled |
| *Rostanga pulchra* | GQ292028 | – | – | GQ326864 | – | Porifera [2] | – | Uniform |
| *Sclerodoris tuberculata* | MF958417 | MF958286 | MF958373 | MF958327 | MN720323 | Porifera [2] | Synthesis [10] | Uniform |
| *Taringa telopia* | MN720291 | KP871700 | – | – | KP871675 | Porifera* [50] (confamilial trait) | – | Stripes/spots, mottled & distinct rim |
| *Thordisa albomacula* | MF958418 | MF958287 | MF958374 | – | – | Porifera [2] | – | Uniform |
| **Dorididae** | | | | | | | | |
| *Aphelodoris* sp. | MF958424 | MF958293 | MF958379 | MF958332 | – | Porifera [2, 51] | – | Mottled |
| *Conualevia alba* | KC153021 | KC153023 | – | – | – | Porifera [52] | – | Uniform |
| *Doriopsis granulosa* | AF249798 | AF249223 | – | AF249212 | – | Porifera [2] | – | Uniform |

*(Continued)*

**Table 1.** (Continued)

| Species | GenBank Accessions | | | | | Prey Group | Chemical Acquisition | Colour Pattern |
|---|---|---|---|---|---|---|---|---|
| | COI | 16S | 28S | 18S | H3 | | | |
| *Doris pseudoargus* | AJ223256 | AJ225180 | – | – | – | Porifera [53] | Synthesis [54-57] | Mottled |
| **Goniodorididae** | | | | | | | | |
| *Ancula gibbosa* | KP340388 | KP340291 | KP340356 | KP340322 | KP340413 | Entoprocta [2, 8, 58-61] | – | Stripes/spots |
| *Goniodoridella savignyi* | OK143202 | – | – | – | – | Bryozoa [3] & Chordata[110] | – | Stripes/spots, mottled & distinct rim |
| *Goniodoris nodosa* | AF249788 | AF249226 | – | AJ224783 | – | Bryozoa[111] & Chordata [53, 62] | Sequestration[63,64] | Mottled & distinct mantle rim |
| *Lophodoris danielsseni* | OK156412 | – | – | – | OK169877 | Bryozoa[112] Chordata (confamilial trait) & Entoprocta [2] | – | Uniform |
| *Murphydoris puncticulata* | OK156427 | – | – | – | OK169891 | Bryozoa [2] & Chordata[112] | – | Stripes/spots, mottled & distinct rim |
| *Okenia vena* | KY661381 | KY661373 | – | – | KY661384 | Bryozoa [3] | Sequestration [10, 65] | Mottled & distinct rim |
| *Trapania reticulata* | MF958432 | MF95803 | – | MF958342 | | Bryozoa [3] & Entoprocta [3, 66, 67] | – | Stripes/spots & mottled |
| **Hexabranchidae** | | | | | | | | |
| *Hexabranchus sanguineus* | MF958433 | MF958305 | MF958388 | MF958344 | – | Porifera [10] | Sequestration [10, 25] | Mottled & distinct rim |
| **Mandeliidae** | | | | | | | | |
| *Mandelia mirocornata* | MF958411 | MF958278 | MF958365 | MF958321 | – | Porifera [68] (based on buccal morphology of sponge-feeding dorids) | – | Stripes/spots |
| **Onchidorididae** | | | | | | | | |
| *Acanthodoris nanaimoensis* | KM219657 | KJ653656 | KP340360 | KP340325 | KM225810 | Bryozoa [2, 70] | Synthesis [10, 71] | Distinct rim |
| *Adalaria slavi* | MN224050 | MN224074 | MN224110 | MN224105 | – | Bryozoa [2, 8, 10, 72, 73] | Sequestration [10] | Uniform |
| *Atalodoris oblonga* | KP340410 | – | KP340385 | KP340349 | KP340430 | Bryozoa [2, 74] | – | Mottled |
| *Onchidoris muricata* | KM219680 | – | KP340383 | KP340348 | KM225830 | Bryozoa [75-78] | – | Mottled |
| *Onchimira cavifera* | MN224073 | MN224104 | MN224137 | MN224109 | – | Bryozoa [73] | – | Uniform |
| **Phyllidiidae** | | | | | | | | |
| *Ceratophyllidia* sp. | MF958413 | MF958281 | MF958368 | MF958323 | – | Porifera* (confamilial trait) | Sequestration* (confamilial trait) | Stripes/spots |
| *Phyllidia coelestis* | MF958412 | MF958279 | MF958366 | – | – | Porifera [9, 79-81] | Sequestration [9, 79-83] | Stripes/spots |
| *Phyllidiella nigra* | – | MF958280 | MF958367 | MF958322 | – | Porifera [84] | Sequestration [85-87] | Stripes/spots |
| *Phyllidiopsis krempfi* | KX235972 | – | – | – | – | Porifera* (based on buccal morphology of sponge-feeding phyllids) | Sequestration [10] | Stripes/spots |
| *Reticulidia halgerda* | MF958414 | MF958282 | MF958369 | – | – | Porifera [2, 88] | Sequestration [10] | Stripes/spots |
| **Polyceridae** | | | | | | | | |
| *Colga pacifica* | MZ782097 | – | – | – | – | – | – | Uniform |
| *Crimora lutea* | EF142903 | EF142950 | – | – | – | Bryozoa [2, 88] | – | Stripes/spots |
| *Gymnodoris ceylonica* | KY806818 | KY806790 | KY806809 | KY806800 | – | Mollusca [89-91] | – | Stripes/spots & distinct rim |
| *Kalinga ornata* | MN224072 | MN224103 | MN224136 | – | – | Generalist [92-94] | – | Stripes/spots |
| *Kaloplocamus* sp. | MF958429 | MF958299 | MF958383 | MF958337 | – | Bryozoa [2, 5] | – | Mottled |

*(Continued)*

**Table 1.** (Continued)

| Species | GenBank Accessions | | | | | Prey Group | Chemical Acquisition | Colour Pattern |
|---|---|---|---|---|---|---|---|---|
| | COI | 16S | 28S | 18S | H3 | | | |
| *Lecithophorus capensis* | MZ382782 | – | – | – | MZ399572 | Bryozoa [2, 95] | – | Uniform |
| *Limacia* sp. | HM162692 | HM162602 | KP340353 | KP340320 | HM162508 | Bryozoa [96, 97] | Synthesis [98] | Stripes/spots |
| *Martadoris mediterranea* | KP793057 | – | – | – | KP793060 | – | – | Mottled |
| *Nembrotha cristata* | MF958431 | MF958301 | MF958385 | MF958339 | | Chordata [10, 25, 99] | Sequestration [10, 25, 100] | Stripes/spots & mottled |
| *Palio dubia* | AJ223272 | AJ225197 | – | – | – | Bryozoa [101] | – | Mottled |
| *Plocamopherus pecoso* | MF958430 | MF958300 | MF958384 | MF958338 | – | Bryozoa [5, 40] | Synthesis [102] | Mottled |
| *Polycera aurantiomarginata* | JX274068 | JX274038 | – | – | – | Bryozoa [2, 103, 104] | Sequestration [204, 205] & synthesis* (some *Polycera* species contain a chemical that is biosynthesized in other nudibranchs) [10, 102] | Stripes/spots & distinct rim |
| *Roboastra gracilis* | EF142863 | EF142912 | – | – | – | Bryozoa [2, 106, 107] | Sequestration [10, 106] | Stripes/spots |
| *Tambja marbellensis* | HM162689 | HM162599 | – | – | HM162505 | Bryozoa [2, 10, 107] | Sequestration [10, 107, 108] | Stripes/spots & distinct rim |
| *Thecacera* sp. | MZ382795 | – | – | – | MZ399586 | Bryozoa [2] | Synthesis [102] | Stripes/spots |
| *Triopha catalinae* | HM162690 | HM162600 | KP340354 | KP340321 | HM162506 | Bryozoa [2, 10, 98, 109] | Sequestration & synthesis [10, 98] | Stripes/spots |
| *Tyrannodoris ernsti* | KJ999212 | – | – | – | KJ999232 | Mollusca [5, 105] | Sequestration [10] | Stripes/spots |
| *Vayssierea* sp. | MZ382796 | – | MF958408 | MF958362 | MZ399587 | Annelida [2, 3, 69] | – | Uniform |
| **Showajidaiidae** | | | | | | | | |
| *Showajidaia sagamiensis* | MN224070 | MN224101 | MN224134 | MN224108 | – | – | – | Stripes/spots |
| **Outgroup** | | | | | | | | |
| *Bathydoris aioca* | KP871635 | KP871682 | – | – | KP871658 | | | |

our three traits using an independent contrasts correlation model in an MCMC framework via a two-step process in BayesTraits v4.1.1 [29]. First, we ran the 'complex' analysis, using a sample period of 1,000 with 1,010,000 iterations and a burn-in of 10,000, and then a stepping stone sampler of 100 stones each run for 1000 iterations, to calculate the covariance between each pair of traits. Then, we ran the 'simple' analysis using the same parameters above but using the TestCorrel command to set the covariance to zero for each pair of traits. Each analysis generated a single log marginal likelihood and these were used to calculate Log Bayes Factors (Log BF) = 2(log marginal likelihood complex model – log marginal likelihood simple model). Log BF values of < 2 are considered weak evidence of correlation [29].

## 3. Results and Discussion

### Is the Doridoidei phylogeny resolved?

Several recent papers have employed multi-marker datasets to reconstruct dorid nudibranch phylogeny with variable results [1,30,31]. Here, we employed IQ-TREE to generate a concatenated ML phylogeny for 88 genera of Doridoidei (Fig 2), with a final alignment length of 3,603 bp. We recovered patterns similar to [1,30] and [31], but with some notable differences. First, we recover Phyllidiidae at the base of the tree with Mandeliidae, but [31] recover

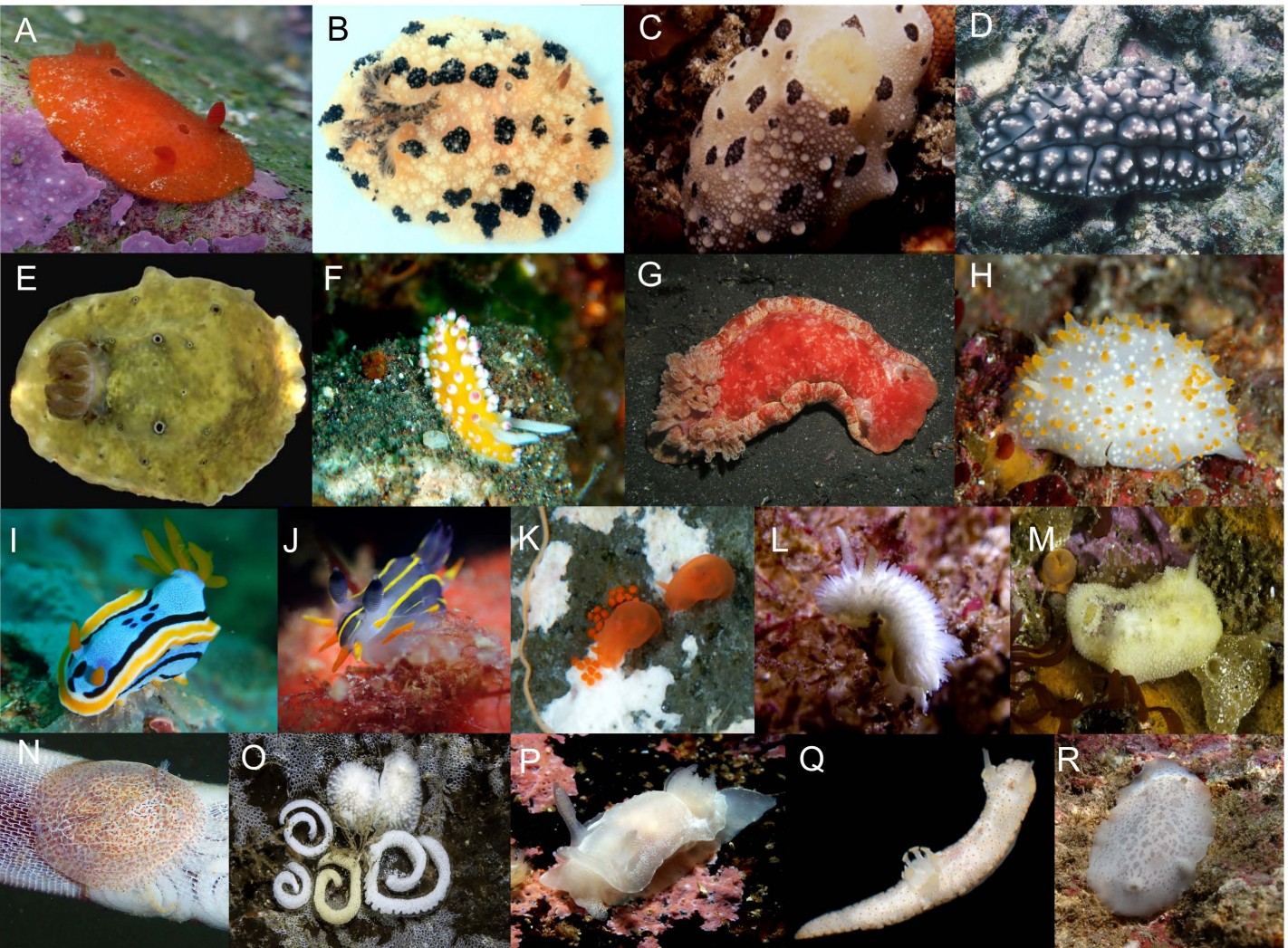

**Fig 1. Representatives of seventeen of the families included in this study, with image attributions in parentheses.** (A) Cadlinidae: *Aldisa* (R. Agarwal), (B) Dendrodorididae: *Dendrodoris* (J. Brodie), (C) Mandeliidae: *Mandelia* (Seascapeza), (D) Phyllidiidae: *Phyllidiopsis* (B. Picton), (E) Actinocyclidae: *Actinocyclus* (B. Rudman), (F) Cadlinellidae: *Cadlinella* (K. Layton), (G) Hexabranchidae: *Hexabranchus* (C. Watanabe), (H) Showajidaiidae: *Showajidaia* (I. Diver), (I) Chromodorididae: *Chromodoris* (K. Layton), (J) Polyceridae: *Polycera* (S. Verheyen), (K) Polyceridae: Okadaiinae: *Vayssierea* (I. Diver), (L) Calycidorididae: *Diaphorodoris* (J. Yasaki), (M) Dorididae: *Doris* (B. Picton), (N) Corambidae: *Corambe* (R Agarwal), (O) Onchidorididae: *Onchidoris* (B. Picton), (P) Goniodorididae: *Goniodoris* (B. Picton), (Q) Aegiridae: *Aegires* (G. Cobb), and (R) Discodorididae: *Discodoris* (P. Bourjon).

Mandeliidae as sister to Aegiridae and [1] do not include Mandeliidae. We also recover a recurring pattern with Dorididae. In our results, Dorididae is polyphyletic because *Aphelodoris* falls outside this family and instead is sister to the Discodorididae + Aegiridae. This is partially consistent with the results from [30], where Dorididae is also polyphyletic due to *Aphelodoris* grouping with Discodorididae, although not with Aegiridae. In contrast, [31], who have dense sampling within Discodorididae, recover Dorididae as monophyletic, but their dataset lacks *Aphelodoris*.

One of the most notable topological differences is the position of Actinocyclidae, which is sister to some, but not all families, in our phylogeny (BS = 84) (i.e., Mandeliidae + Phyllidiidae, Dendrodorididae and Cadlinidae are at the base of the tree), similar to [31]. Conversely, some earlier results recover Actinocyclidae as sister to all other Doridoidei families [1,30]. The

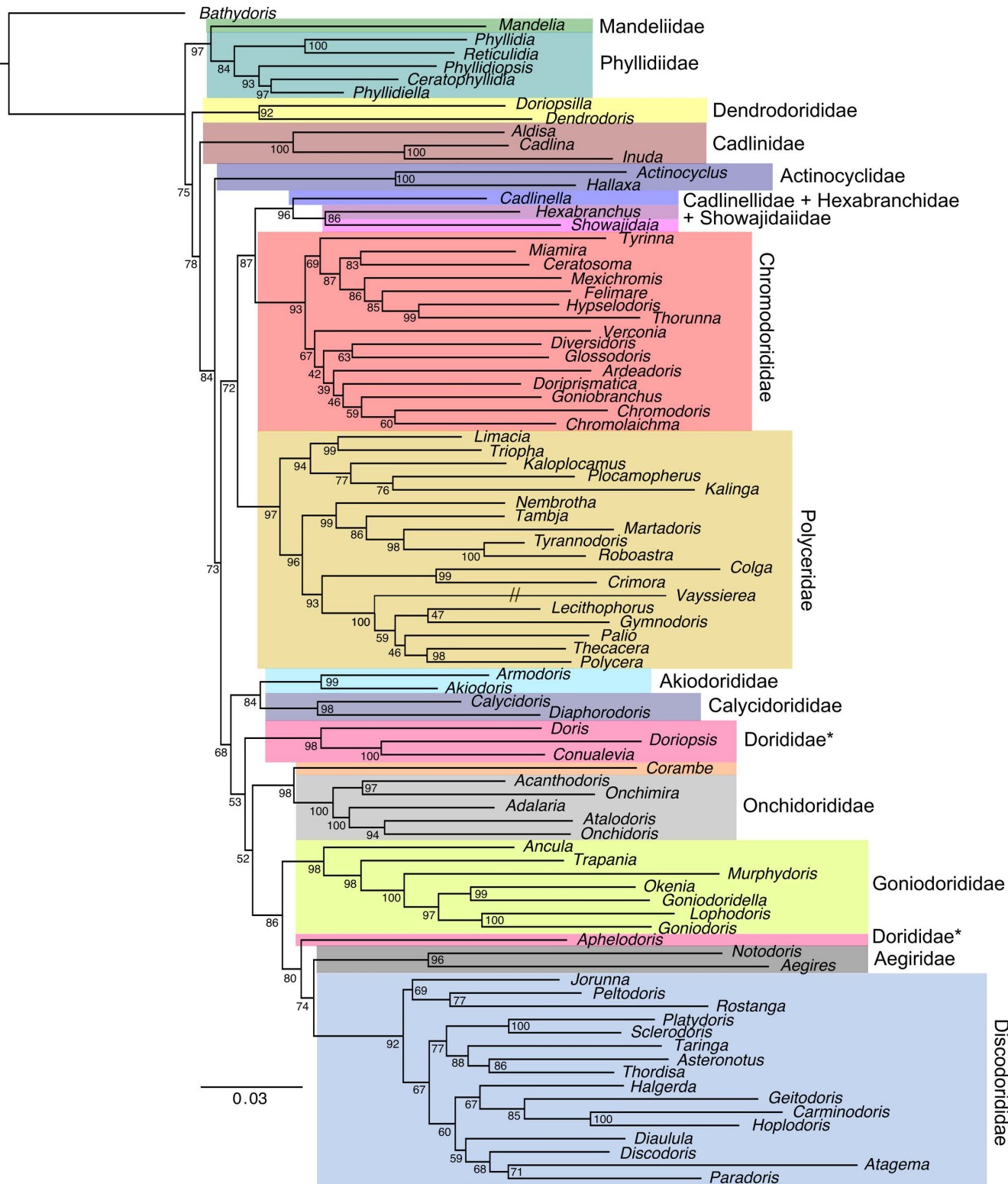

**Fig 2. Multi-gene phylogeny (COI,16S,18S,28S,H3) for** Doridoidei**, constructed with IQ-TREE (1000 UF bootstraps, GTR + I + G) with families denoted by coloured boxes and bootstrap support values provided at each node.** Non-monophyletic families are marked with an asterisk. Hash marks denote that the branch has been truncated to one half of its original length.

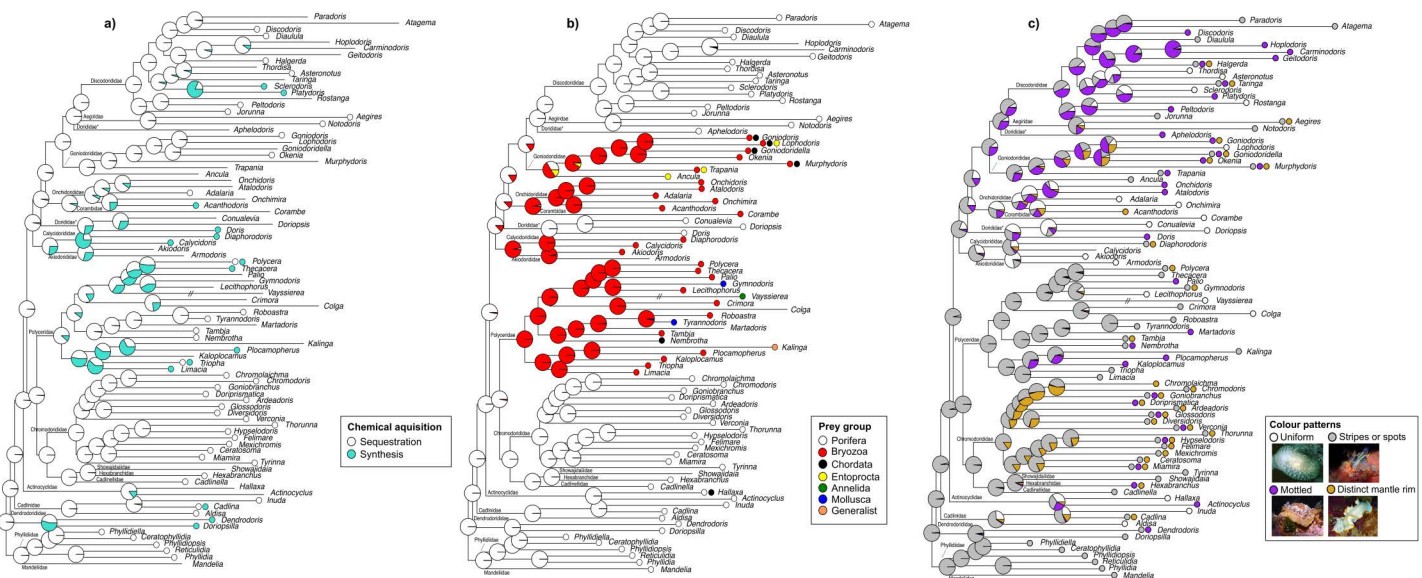

**Fig 3. Ancestral state reconstructions of (a) chemical acquisition, (b) prey preference and (c) colour patterns in Doridoidei, run with an equal rates (ER) model in corHMM in R.** Pie charts at each node represent marginal likelihoods of ancestral states. Polymorphic character states are marked with two or more data points in extant taxa which are labeled with genus names. Taxa without a data point are missing the relevant trait data. Family names are provided at the relevant node and non-monophyletic families are marked with an asterisk. Hash marks denote that the branch has been truncated to one half of its original length. Inset in (c) shows examples of each colour pattern. From left to right, top to bottom, with image attributions in parentheses: *Adalaria* (B. Picton), *Polycera* (S. Verheyen), *Hoplodoris* (S. Rohrlach) and *Ardeadoris* (S. Graham).

position of Discodorididae also varies tremendously, with Discodorididae sister to Aegiridae in our study, albeit with weak support (BS = 74), compared to earlier results that show Discodorididae sister to Dorididae [30] or Goniodorididae [1]. Additionally, the clade containing Cadlinellidae, Hexabranchidae and Showajidaiidae is sister to Chromodorididae in our phylogeny (BS = 87) but sister to Polyceridae in [30], unresolved in [1] and sister to both Chromodorididae and Polyceridae in [31]. We also conducted a Bayesian analysis (S1 Fig) but many deeper nodes (i.e., relationships among families) were poorly supported and subsequently collapsed into polytomies. Despite this poor resolution, we still recover similar family-level relationships as in our ML analysis, except that Polyceridae is non-monophyletic. Another difference between the ML and Bayesian analysis is the position of Goniodorididae which appears as sister to a clade containing Aegiridae and Discodorididae in the former but as sister to the rest of the Doridoidei in the latter. However, the result from the Bayesian analysis has never been recovered in any molecular or morphological phylogenies to date. This significant topological variability among methods and studies likely reflects differences in taxon and gene sampling as well as phylogenetic methodology. For instance, [1] and [30] include 56 genera while here we include all 88 genera for which genetic data is currently available. Additionally, [1,30] and [31] employ three or four gene datasets, spanning a total of five unique genes, for which we incorporate data from all five loci here, however with variable completeness across our dataset. In any case, we report multiple instances of poor support within Doridoidei (i.e., BS < 95) and these relationships should be interpreted with caution until additional genomic data becomes available for phylogenetic reconstruction.

Due to the many examples of poor support across Doridoidei, future work should employ reduced representation or whole genome approaches to generate larger datasets that have greater power for phylogenetic resolution. In fact, transcriptomes [32,33], exons [34], ultra-conserved elements (UCEs) [35] and mitogenomes [36] have already been used to resolve

both deep and shallow evolutionary relationships in heterobranchs, but these studies did not target Doridoidei specifically. With respect to taxon sampling, we restricted our dataset to just a single representative per genus and while this might contribute to topological uncertainty, the lack of a densely sampled phylogeny across these same genera limits this investigation. One oddity is the long branch leading to *Vayssierea*, an intertidal nudibranch genus currently residing in the Okadaiinae subfamily within Polyceridae. This pattern could reflect either undersampling of this subfamily or accelerated evolutionary rates, the latter of which has recently been uncovered in chromodorid nudibranchs [37]. Lastly, we employ IQ-TREE for ML analysis while previous studies mostly employed RAxML [38], which may contribute to the topological variation observed here. In all, there remain issues with poor support across the Doridoidei phylogeny that is unlikely to be resolved without additional sampling, both taxonomic and genomic. Even with denser sampling, we may continue to face challenges when reconstructing dorid nudibranch phylogeny.

## How did chemical defence, prey preference and colour patterns evolve in Doridoidei?

Ancestral state reconstruction recovered the most recent common ancestor (MRCA) of all Doridoidei as a sponge-feeder with complex colour patterns (e.g., stripes or spots) that sequestered chemicals from its prey (Fig 3a–3c). Looking at chemical acquisition, *de novo* synthesis has evolved multiple times independently from a sequestering ancestor (Fig 3a). Interestingly, members of Bathydoridoidei (the sister group to Doridoidei) tend to *de novo* synthesize and thus a switch to a putatively less costly mechanism, like sequestration (e.g., [39]), may also have facilitated the diversification of Doridoidei. Within Doridoidei, both modes of chemical acquisition are reported from sponge and bryozoan feeders. Entoproct feeding taxa are reported as sequesterers in the literature. Little is known about secondary metabolites in this unique prey group and thus the taxa reported here as entoproct feeders may in fact feed on and sequester metabolites from a more diverse suite of prey. We found little variation in chemical acquisition at the genus level, with the exception of *Polycera* and *Cadlina* where different species, and even populations of the same species, have been shown to both sequester and synthesize (e.g., [40]). We expected to find that prey switching evolved concurrently with switches to different chemical acquisition modes, but there is little support for this. However, it is important to note that data on chemical acquisition in nudibranchs is patchy and, in some cases, may be speculative. As such, the results presented here should be interpreted with caution until more robust data is available for re-investigation.

Looking at prey preference, feeding on bryozoans most likely evolved four times independently from a sponge feeding ancestor at relatively deep nodes of the phylogeny (Fig 3b). This includes in the MRCA of 1) Polyceridae, 2) Akiodorididae and Calycidorididae, 3) Onchidorididae and Corambidae, and 4) Goniodorididae. However, prey preference is complex in Goniodorididae and thus it is possible that the MRCA fed on entoprocts, bryozoans and other diverse taxa. Feeding on chordates (tunicates) evolved at least three times, from both sponge and bryozoan feeding ancestors. Feeding on molluscs has evolved twice, only within the Polyceridae, from a bryozoan feeding ancestor. Conversely, entoproct feeding may have evolved multiple times within Goniodorididae, or once in the MRCA and then subsequently lost in some taxa, being replaced with bryozoan and tunicate feeding. Alternatively, it is possible that goniodorids are feeding on entoprocts that are found amongst bryozoans and tunicates and in fact this feeding preference may be more prevalent than reported in the literature. Interestingly, sponge feeding was not re-gained in any of the lineages where prey-switching occurred. Most genera feed on specific prey groups, with the exception of

most goniodorids that feed on two types of organisms (bryozoans and tunicates or tunicates and entoprocts), the generalist genus *Kalinga* and the genus *Hallaxa* (sponges and ascidians). These findings align, in part, with previous work that investigated the evolution of prey preference in cladobranch slugs and uncovered widespread hydrozoan feeding that originated in the MRCA with subsequent switches to specific prey groups [5]. Additionally, Bathydoridoidei, the sister clade to Doroidei, is a much less diverse taxonomic group that tends to exhibit generalist feeding behaviour [10]. As such, the switch to sponge-feeding in the MRCA of Doridoidei may have facilitated their impressive diversification.

Nudibranch colour patterns are complex and their use in signaling may be conditional or vary across different ecological contexts (i.e., what might be considered aposematic in one environment is cryptic in another). Nonetheless, we recover the MRCA of all Doridoidei as having spots or stripes, suggesting that the ancestor displayed complex patterns and that more cryptic patterns subsequently evolved in multiple lineages (Fig 3c). Given the complexity of these colour patterns (i.e., multiple possible colour patterns are present within a single genus) it is also likely that the MRCA(s) exhibited a combination of these patterns, and it is likely that the full spread of colour pattern variation within genera is not captured here. Nonetheless, previous work also recovered a spotted phenotype in the MRCA of a clade of weevils that display diverse colour patterns ranging from uniform to complex net-like patterns [41], but in contrast, complex colour patterns evolved from a uniform ancestor in aposematic beetles [42]. Looking across dorid families, both Chromodorididae and Goniodorididae appear to have the most complex and variable colour patterns, where stripes/spots, mottled patterns and distinct mantle rims appear within many genera. In fact, a distinct mantle rim was most common in Chromodorididae and Goniodorididae, and evolved from spotted/striped, mottled and even uniform ancestors, but it was absent in Phyllidiidae, Akiodorididae, Calycidorididae, Dorididae and Corambidae. Conversely, uniform species were most common in the Akiodorididae, Calycidorididae, Dorididae, Corambidae, and Onchidorididae, and absent from the Chromodorididae. Colour pattern was only consistent in a single family, with all members of the Phyllidiidae exhibiting spots or stripes. The mottled colour pattern occurs multiple times across diverse taxonomic groups so it's likely that this pattern is conditional in that it ranges from aposematic to cryptic depending on the context. Given the link between diet, chemical defence and colour patterns (e.g., [11,43]), a signal of correlated evolution amongst these three traits might be expected, however, we find weak evidence of this in our BayesTraits analysis (LogBF = 1.73). The lack of evidence for correlated evolution may reflect missing trait data or the lack of fine-scale resolution and thus future work should look to investigate the evolution of these traits across a species-level phylogeny to elucidate clearer patterns.

## 4. Conclusions

Here, for the first time, we investigate, in tandem, the evolution of prey preference, chemical acquisition and colour pattern in dorid nudibranchs. We use previously published genetic and trait data to support this work, demonstrating the importance of existing databases in advancing our understanding of evolution in non-model, and typically data-deficient, organisms. Despite these advances, there are some key limitations of this work. First, the dorid phylogeny employed for ancestral state reconstruction remains partially unresolved and thus a more comprehensive, species-level phylogeny is needed to confirm the patterns shown here. Additionally, much of the trait data retrieved from the literature is based on just a few representative species per genus or family and thus it is likely that these traits vary both within and among genera and our results might instead reflect broader patterns. Future work should look to revisit these findings and patterns with a fully resolved phylogeny and refined trait data, although more work is needed to improve our understanding of basic biology and ecology in

these organisms to support this. Nonetheless, limiting our work to well-sampled and well-known taxonomic groups will only continue to contribute to the pronounced taxonomic bias in evolutionary biology.

## Supporting information

**S1 Fig. Multi-gene Bayesian phylogeny (COI,16S,18S,28S,H3) for Doridoidei with posterior probabilities provided at each node. Hash marks denote that the branch has been truncated to one half of its original length.**
(TIF)

**S1 File. Supplementary references (Table 1).**
(DOCX)

## Acknowledgements

We thank two anonymous reviewers for their helpful feedback on this manuscript.

## Author contributions

**Conceptualization:** Kara K. S. Layton.

**Data curation:** Silvia Prieto-Baños, Kara K. S. Layton.

**Formal analysis:** Silvia Prieto-Baños, Kara K. S. Layton.

**Investigation:** Silvia Prieto-Baños, Kara K. S. Layton.

**Methodology:** Silvia Prieto-Baños, Kara K. S. Layton.

**Project administration:** Kara K. S. Layton.

**Supervision:** Kara K. S. Layton.

**Visualization:** Silvia Prieto-Baños, Kara K. S. Layton.

**Writing – original draft:** Silvia Prieto-Baños, Kara K. S. Layton.

**Writing – review & editing:** Silvia Prieto-Baños, Kara K. S. Layton.

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
