## [Decision Letter · Decision Letter 0]

18 Jun 2024

PONE-D-24-20562Tracing the evolution of key traits in dorid nudibranchsPLOS ONE

Dear Dr. Layton,

Thank you for submitting your manuscript to PLOS ONE. After careful consideration, we feel that it has merit but does not fully meet PLOS ONE’s publication criteria as it currently stands. Therefore, we invite you to submit a revised version of the manuscript that addresses the points raised during the review process.

We look forward to receiving your revised manuscript.

Kind regards,

Satheesh Sathianeson, Ph.D

Academic Editor

PLOS ONE

Journal Requirements:

Reviewers' comments:

Reviewer's Responses to Questions

**Comments to the Author**

1. Is the manuscript technically sound, and do the data support the conclusions?

Reviewer #1: Yes

Reviewer #2: Partly

2. Has the statistical analysis been performed appropriately and rigorously? 

Reviewer #1: Yes

Reviewer #2: I Don't Know

3. Have the authors made all data underlying the findings in their manuscript fully available?

Reviewer #1: Yes

Reviewer #2: Yes

4. Is the manuscript presented in an intelligible fashion and written in standard English?

Reviewer #1: Yes

Reviewer #2: Yes

5. Review Comments to the Author

Reviewer #1: In the manuscript, the authors traced the evolution of key traits in dorid nudibranchs using accumulated data. The study provides insight into the most recent common ancestor of the nudibranch group, a complicated subject because of the lack of fossil records. Therefore, the manuscript is potentially considered for publication in the journal. However, there are some points in the manuscript that the authors should check and improve before the final decision of the editor.

- The first thing the authors should notice is the novelty of the study. Actually, the findings of the manuscript could be found elsewhere in the previous study. Thefore, the authors should add more information to highlight their novel points in the study.

- The findings of the current study mainly relied on the phylogeny of dorid nudiranch. However, it should be noted that the investigation of dorid nudibranch phylogeny is a complicated subject as tree topology is varied if different markers or approaches are used. Therefore, in addition to a few genes, larger data such as the mitogenome and transcriptome have been used for nudibranchs and may provide better resolutions. Some studies have been published, and the authors should refer to the manuscript. Also, in tree construction, different approaches are used, such as maximum likelihood and Bayesian inference. Therefore, the authors could consider adding a tree built by Bayesian inference to see similarities and differences between the two methods.

- As the author stated in the manuscript, “much of the trait data employed here lacks species-level resolution and might instead reflect broader patterns.” One of the key traits of the dorid nudibranch is the structure of the gill and gill cavity. For these characteristics, Korshunova et al. (2020) have already discussed them in detail. Even though in the present study, the authors did not include the gill and gill cavity for analyses, discussion about these in relation to the phylogeny of the dorid nudibranch is necessary.

- Some citations are missing from the reference, such as “Hallas et al. 2017." The authors should carefully check and include them.

Reviewer #2: This manuscript describes the authors’ use of publicly available data to construct a phylogeny and trace the evolution of feeding, color pattern and chemical defense (whether sequestered, secondarily modified or synthesized de novo) within dorid nudibranchs. Overall, it is nice to see publicly available data being used to tackle interesting questions about the evolution of color patterns in nudibranchs, which is an area very much in its infancy. However, there are certainly limitations to using these data within dorids since to date, no previously published phylogenetic analysis of Sanger-generated markers has successfully resolved dorid phylogeny (and this problem has yet to be resolved—if possible—with high throughput DNA sequencing data). The lack of phylogenetic resolution seems to be a huge hurdle to tracing evolution of traits within this group. Based on their analyses, the authors claim that the most recent common ancestor to dorids likely fed on sponges, possessed complex color patterns, and sequestered defensive metabolites from their sponge prey. In it’s present form, this manuscript is not ready for publication, but I encourage the authors to revise and resubmit. Generally, there are some definitions that need to be better defined with respect to the traits explored in the paper, and more details need to be included for the methods and results. There are also parts of the paper that are factually inaccurate and need correction.

I have included some general suggestions for the authors on ways to improve the manuscript, as well as more specific suggestions according to specific line in the manuscript.

Some thoughts on chemical defense: Chemical defense within sea slugs is not my primary of expertise, but it is unclear to me whether there is a distinction to be made between the use of secondary metabolites vs acid secretion within the context of heterobranch “chemical defense”. I have interpreted this distinction from existing literature. Are both of these considered chemical defense in the context of this study? I think the former are far more common within nudibranchs, and the latter far more common in the sister taxon, Pleurobranchomorpha (see Wägele et al., 2017), but acid secretion is used for defense in at least some nudibranchs. However, perhaps this is an outdated or oversimplified artificial distinction (?). Are the two phenomena related in some way, e.g. developmentally? If there is a distinction to be made, then I think it would make the paper more robust to make/address that distinction in the introduction or discussion. In other words, how chemical defense is defined will have an impact on our understanding of how the mechanisms have evolved within sea slugs. Similarly, does secondary modification imply that species that do this are also sequestering from their prey? If so, then shouldn’t the taxa that are considered to be secondary modifiers also sequesters? If a species synthesizes de novo, does that mean that by definition they are incapable of secondary modification? or can they be considered to “secondarily modify” if they modify a molecule that they synthesized themselves? It would better serve this work to clearly define what each of these mean.

Lines 36-7 contain what appears to be an error, stating that Doridoidei “is the most diverse heterobranch group.” This depends on what the author means by “diverse”. The most speciose heterobranch group is most definitely the terrestrial Stylommatophora, but perhaps the author is referring to diversity of morphology? or perhaps meant to say the most diverse/ or speciose marine heterobranch group. There are many ways to resolve this.

Line 44: I “species poor” might be a better alternative to “less-diverse” sister group

Lines 46-47: There are so many species of dorids that do not feed on sponges. Is there a citation to support that most are thought to feed exclusively on sponges? or some basic numbers you could include here? As written this feels like an oversimplification. I can think of many dorids that feed on bryozoans, barnacles, tunicates, spirorbid worms, entoprocts, even ophiuroids. Perhaps a better distinction to be made here, which I think you started to do is generalist feeders vs specialists.

Line 48: I suggest to add “marine” in front of “heterobranchs” or find a citation that suggests this is applicable for terrestrial heterobranchs (if it is!). Terrestrial heterobranchs vastly outnumber marine ones, but these two citations apply to marine lineages.

Line 68: substitute “often” for “typically” and is there a reference you can cite here?

Line 75-77: These citations focus on chromodoridids, Knutson and Gosliner 2022 is another, more recently published example of distantly related species sharing similar color patterns, but from within a group of non sponge feeding, non-chromodoridid dorids

Line 84: Presumably this has been studied within marine fishes, which are speciose and also feature many diverse color patterns? If so, then this should be touched upon here, or if not, then this could be used as an extra point for how little is known in marine systems, otherwise, specify marine invertebrates

Lines 85-86: Perhaps using molecular methods, but authors have previously discussed evolutionary scenarios for evolution of prey and metabolite sequestration vs synthesis de novo, see for example Cimino and Ghiselin 1999 and references therein. It would be good/appropriate to acknowledge this historical context since this has been a question of interest for some time.

Methods: “Mining” seems vague here. “Data mining” often implies computer automation, but if you automated this process, then there are details that are missing from within the methods here. I suggest modifying the language with to language without misleading connotations

Lines 104-5: Did you encounter multiple representatives that may have had “sufficient genetic data to support phylo. reconstruction”? if so, how did you choose which taxa/sequences to use?

Line 125: brackets- in my copy they are parentheses, not brackets

Table 1: While a supplementary document is included that lists the references used to complete the trait data in this table, it is not clear in many cases which reference was used to specify trait data for a particular taxon. It is essential that the authors link the specific traits to the actual sources for the work to be fully transparent and useful to future researchers—perhaps this can be done with superscripts or subscripts, if allowed by the journal formatting.

Phylogenetic and ancestral state reconstruction.

The authors chose to only include one phylogenetic reconstruction method (maximum likelihood, ML with bootstraps) for their phylogeny, though it is generally accepted within systematic studies to additionally run Bayesian Inference and present the support values from each method. Occasionally these methods corroborate each other and other times they may show conflict. Given the lack of resolution of this phylogeny, it seems particularly appropriate to include results from Bayesian Inference. Authors should perform Bayesian Inference, ensuring convergence of chains, and present these results alongside the ML results, addressing any conflicts or consistencies between methods.

I have minimal experience with Ancestral State Reconstruction, so I cannot directly comment on the methodology employed by the authors for this purpose.

Lines 161/2: Again, I’m less not very familiar with the details of using BayesTraits, but certainly some details of the analyses have been left out of this section? MCMC should have parameters that are set, right? and you need to have reached convergence for the results to mean anything… I couldn’t find it mentioned in the paper that convergence was reached for this analysis.

Line 167: replace “resolve” with “reconstruct”

Line 176: Please cite here the studies that find Actinocyclidae as sister to all other Doridoidei. E.g. Kurshonova et al 2020 does not demonstrate that Actinocyclidae is the sister group to the rest of Doridoroidei. If you look closely at their tree, most nodes at the backbone of dorids have no support (they neither listed the actual support, nor collapsed the unsupported nodes).

The Hallas paper presented many differing topologies, in part to show how different alignment methods impacted the resulting trees. What is more interesting, is that this lack of phylogenetic resolution at the backbone of dorids likely reflects rapid diversification and ancient incomplete lineage sorting, etc that makes this phylogeny particularly difficult to resolve.

Lines 184/5: To be fair, some of these genera have been described or resurrected since these two older papers were published.

Line 235-238: It is not clear what this sentence or the reference is meant to add here to this section. please rework

Line 245: Technically, the species of Gymnodoris that you’ve included in your phylogeny (G. ceylonica) does not feed on nudibranchs, it is documented to feed on other lineages of sea slugs. Remove “(nudibranchs)”

Line 283: This sentence contradicts your dataset. Table 1 shows that Lecithophorus, Colga and Vayssierea (which, less face it…it’s a polycerid) are all uniform in color pattern, and I can think of other polycerids as well. “entirely absent” is not true in the case of polycerids.

References: The authors must double check the entire manuscript to ensure that all references cited in the text appear in this section. For example, I noticed that Hallas et al. 2017 was cited a few times, but it is not listed in this section.

Figures: The full tif file of images (Fig 1) looks great. However, the downsampled image that downloaded as part of the reviewer pdf is way too dark and compromises the messages that the authors are trying to make about color patterns. To the authors- make sure to work with the editors to make sure that the final published version is still visually acceptable

6. PLOS authors have the option to publish the peer review history of their article (what does this mean? ). If published, this will include your full peer review and any attached files.

**Do you want your identity to be public for this peer review?** For information about this choice, including consent withdrawal, please see our Privacy Policy .

Reviewer #1: No

Reviewer #2: No

---

## [Author Response · Author response to Decision Letter 1]

13 Dec 2024

Response to Reviewers

We thank both reviewers for their helpful revisionary suggestions and comments that have significantly improved this manuscript. We provide a detailed response to each of the reviewer’s comments below and have made the corresponding changes throughout the manuscript.

Reviewer One

C1_1: The first thing the authors should notice is the novelty of the study. Actually, the findings of the manuscript could be found elsewhere in the previous study. Therefore, the authors should add more information to highlight their novel points in the study.

R1_1: We thank the reviewer for their concerns about novelty, and we certainly want to make sure that we’re communicating the novelty and importance of this work. We agree, in part, that a multi-gene phylogeny for dorids has been presented in earlier papers (both by Korshunova et al. 2020 and Hallas et al. 2017), but in fact, we do add new data (both taxa and loci) to this phylogeny. We would argue, however, that the most significant and novel elements of this work are 1) the compilation of one of the most extensive trait datasets to date for dorid nudibranchs and 2) a first exploration of the evolution of these traits over deep time with insights into evidence (or lack thereof) of correlated evolution amongst these traits. It is important for the reviewer to know that this work stems from an MSc project that took place during the pandemic where we were unable to generate new data. I hope the past few years have shown us (the scientific community) the power of compiling and analyzing existing data in new and exciting ways.

C1_2: The findings of the current study mainly relied on the phylogeny of dorid nudibranch. However, it should be noted that the investigation of dorid nudibranch phylogeny is a complicated subject as tree topology is varied if different markers or approaches are used. Therefore, in addition to a few genes, larger data such as the mitogenome and transcriptome have been used for nudibranchs and may provide better resolutions. Some studies have been published, and the authors should refer to the manuscript. Also, in tree construction, different approaches are used, such as maximum likelihood and Bayesian inference. Therefore, the authors could consider adding a tree built by Bayesian inference to see similarities and differences between the two methods.

R1_2: We agree with the reviewer that genome-wide data will be absolutely critical for resolving nudibranch phylogeny (if at all). We do discuss recent efforts to sequence exons, ultraconserved elements (UCEs) and whole transcriptomes on lines 215-224 of the combined Results and Discussion. We did however overlook the inclusion of recent mitogenome work and one transcriptome paper, which we now include in this section. We also now include the Bayesian phylogeny as this was also suggested by Reviewer 2, please see Figure S1 and lines 156-158 and 200-207 in the main text.

C1_3: As the author stated in the manuscript, “much of the trait data employed here lacks species-level resolution and might instead reflect broader patterns.” One of the key traits of the dorid nudibranch is the structure of the gill and gill cavity. For these characteristics, Korshunova et al. (2020) have already discussed them in detail. Even though in the present study, the authors did not include the gill and gill cavity for analyses, discussion about these in relation to the phylogeny of the dorid nudibranch is necessary.

R1_3: We thank the authors for their comment, and we appreciate the wonderful body of morphological work that was presented in Korshunova et al. 2020. We have not discussed the gill and gill cavity work here since it doesn’t explicitly link to our aims/objectives in trying to understand the intimate link between prey preference, chemical defence and colour pattern variation. It seems logical to link these traits given that both metabolites (chemicals) and pigments (colour) can be sequestered from prey, but we don’t believe that gill morphology is a relevant trait to include here. Again, we absolutely agree that the paper by Korshunova et al. 2020 provides an incredibly comprehensive overview of gill and gill cavity morphology and evolution, and we commend these authors for their work, but we don’t believe this needs reiterating in our manuscript.

C1_4: Some citations are missing from the reference, such as “Hallas et al. 2017." The authors should carefully check and include them.

R1_4: Thank you for noticing this. We have since carefully revised the references.

Reviewer Two

C2_1: Some thoughts on chemical defense: Chemical defense within sea slugs is not my primary of expertise, but it is unclear to me whether there is a distinction to be made between the use of secondary metabolites vs acid secretion within the context of heterobranch “chemical defense”. I have interpreted this distinction from existing literature. Are both of these considered chemical defense in the context of this study? I think the former are far more common within nudibranchs, and the latter far more common in the sister taxon, Pleurobranchomorpha (see Wägele et al., 2017), but acid secretion is used for defense in at least some nudibranchs. However, perhaps this is an outdated or oversimplified artificial distinction (?). Are the two phenomena related in some way, e.g. developmentally? If there is a distinction to be made, then I think it would make the paper more robust to make/address that distinction in the introduction or discussion. In other words, how chemical defense is defined will have an impact on our understanding of how the mechanisms have evolved within sea slugs. Similarly, does secondary modification imply that species that do this are also sequestering from their prey? If so, then shouldn’t the taxa that are considered to be secondary modifiers also sequesters? If a species synthesizes de novo, does that mean that by definition they are incapable of secondary modification? or can they be considered to “secondarily modify” if they modify a molecule that they synthesized themselves? It would better serve this work to clearly define what each of these mean.

R2_1: We thank the reviewer for providing this important insight about nudibranch chemical defence. In light of this comment, we have simplified the trait data to include only ‘Sequester’ or ‘Synthesize’. Our motivation for making this change is, in part, because after having re-reviewed the literature and our data we simply don’t have enough information to confidently assign ‘Secondary Modification’ to many groups. We do have this information for some (the 6 taxa which were marked as such in the first version of the manuscript) but that information is lacking for others, and as the reviewer points out, it’s possible that even those taxa that de novo synthesize still secondarily modify their molecules. We have updated the trait data and corresponding figures (Figure 3) accordingly (and see lines 117-119). We appreciate the comment about acid secretion, although we don’t feel there’s enough information available to make this distinction here. The prevailing mechanism is the use of secondary metabolites.

C2_2: Lines 36-7 contain what appears to be an error, stating that Doridoidei “is the most diverse heterobranch group.” This depends on what the author means by “diverse”. The most speciose heterobranch group is most definitely the terrestrial Stylommatophora, but perhaps the author is referring to diversity of morphology? or perhaps meant to say the most diverse/ or speciose marine heterobranch group. There are many ways to resolve this.

R2_2: We thank the reviewer for noticing this oversight and we have revised the text on lines 36-38 to read:

“The Doridoidei infraorder (herein referred to as dorids) is one of the most speciose nudibranch groups, comprising over 2,000 species”

C2_3: Line 44: “species poor” might be a better alternative to “less-diverse” sister group

R2_3: We have revised the text on lines 44-45 to read:

“For instance, Bathydoridoidei, the comparably species poor sister group to Doridoidei“

C2_4: Lines 46-47: There are so many species of dorids that do not feed on sponges. Is there a citation to support that most are thought to feed exclusively on sponges? or some basic numbers you could include here? As written this feels like an oversimplification. I can think of many dorids that feed on bryozoans, barnacles, tunicates, spirorbid worms, entoprocts, even ophiuroids. Perhaps a better distinction to be made here, which I think you started to do is generalist feeders vs specialists.

R2_4: We thank the reviewer for this suggestion and we agree that our original statement wasn’t accurate. We have since revised the text below on lines 46-47 and include a reference to the original paper that proposed this specialization:

“...while most members of the Doridoidei are thought to have more specialized diets (Faulker & Ghiselin 1983)”

C2_5: Line 48: I suggest to add “marine” in front of “heterobranchs” or find a citation that suggests this is applicable for terrestrial heterobranchs (if it is!). Terrestrial heterobranchs vastly outnumber marine ones, but these two citations apply to marine lineages.

R2_5: We have the revised the text to specify “marine heterobranchs” as suggested.

C2_6: Line 68: substitute “often” for “typically” and is there a reference you can cite here?

R2_6: We have made this substitution and added a reference here.

C2_7: Line 75-77: These citations focus on chromodoridids, Knutson and Gosliner 2022 is another, more recently published example of distantly related species sharing similar color patterns, but from within a group of non sponge feeding, non-chromodoridid dorids

R2_7: We thank the reviewer for this helpful suggestion, we have now added Knutson and Gosliner 2022 on line 77.

C2_8: Line 84: Presumably this has been studied within marine fishes, which are speciose and also feature many diverse color patterns? If so, then this should be touched upon here, or if not, then this could be used as an extra point for how little is known in marine systems, otherwise, specify marine invertebrates

R2_8: We thank the reviewer for this helpful suggestion. There hasn’t been much work, if any, linking these traits in marine fishes either (like the polychromatic and venomous blennies, for example), but we have modified this sentence to satisfy the next comment from the reviewer. Please see below.

C2_9: Lines 85-86: Perhaps using molecular methods, but authors have previously discussed evolutionary scenarios for evolution of prey and metabolite sequestration vs synthesis de novo, see for example Cimino and Ghiselin 1999 and references therein. It would be good/appropriate to acknowledge this historical context since this has been a question of interest for some time.

R2_9: We appreciate this feedback and agree that this sentence needed revising. We have since revised as follows on lines 85-87:

“Prey preference, chemical acquisition and colour pattern have been described and studied across several nudibranch groups (see above), but no study has investigated the evolution of these traits in tandem, especially in a phylogenetic context.”

C2_10: Methods: “Mining” seems vague here. “Data mining” often implies computer automation, but if you automated this process, then there are details that are missing from within the methods here. I suggest modifying the language with to language without misleading connotations

R2_10: We generally disagree with this comment from the reviewer and have used, and seen others use, ‘mining’ in a manual context. That being said, it’s possible that others may misinterpret its usage as well and thus we substitute ‘compiling’ for ‘mining’ on line 102.

C2_11: Lines 104-5: Did you encounter multiple representatives that may have had “sufficient genetic data to support phylo. reconstruction”? if so, how did you choose which taxa/sequences to use?

R2_11: Yes there were multiple individuals per species and also multiple species per genus available on GenBank, however, we chose those that had the best genetic coverage. For example, if there were two species of Goniobranchus on GenBank but one of these species had only two loci available, then we prioritized the species that had three or more loci. We felt it was critically important to have as complete a genetic matrix as possible for this Doridina phylogeny.

C2_12: Line 125: brackets- in my copy they are parentheses, not brackets

R2_12: This has been changed to parentheses on line 148.

C2_13: Table 1: While a supplementary document is included that lists the references used to complete the trait data in this table, it is not clear in many cases which reference was used to specify trait data for a particular taxon. It is essential that the authors link the specific traits to the actual sources for the work to be fully transparent and useful to future researchers—perhaps this can be done with superscripts or subscripts, if allowed by the journal formatting.

R2_13: We thank the reviewer for this helpful suggestion and we have implemented these changes by linking specific trait data with corresponding references with superscripts in the table. Please see revised Table 1 and Supplementary File 1. Note, we also refined Table 1 to more explicitly highlight where trait data derives from family-level patterns, when we felt confident to do so (i.e. when there was no variation in traits across the rest of the confamilials and/or with accompanying morphological evidence).

C2_14: Phylogenetic and ancestral state reconstruction.

The authors chose to only include one phylogenetic reconstruction method (maximum likelihood, ML with bootstraps) for their phylogeny, though it is generally accepted within systematic studies to additionally run Bayesian Inference and present the support values from each method. Occasionally these methods corroborate each other and other times they may show conflict. Given the lack of resolution of this phylogeny, it seems particularly appropriate to include results from Bayesian Inference. Authors should perform Bayesian Inference, ensuring convergence of chains, and present these results alongside the ML results, addressing any conflicts or consistencies between methods.

R2_14: We thank the reviewer for this suggestion. We have since implemented a Bayesian analysis in MrBayes and we include a figure in the supplementary materials (see Fig S1). The Bayesian tree has poor support at interior nodes and thus very few relationships amongst families are resolved. However, the same familial relationships are recovered in the Bayesian analysis as in the ML analysis reported in the main text. We include the relevant methods and results for this analysis in the manuscript, on lines 156-158 and 200-207.

C2_15: I have minimal experience with Ancestral State Reconstruction, so I cannot directly comment on the methodology employed by the authors for this purpose.

R2_15: The method employed here, using the corHMM package in R, was chosen because it allows you to test both equal rates and all-rates-different models, and it allows you to incorporate both missing data and multistate traits.

C2_16: Lines 161/2: Again, I’m not very familiar with the details of using BayesTraits, but certainly some details of the analyses have been left out of this section? MCMC should have parameters that are set, right? and you need to have reached convergence for the results to mean anything… I couldn’t find it mentioned in the paper that convergence was reached for this analysis.

R2_16: We thank the reviewer for noticing this as we did forget to specify the BayesTraits parameters, and the iterative process of this analysis, in the original version of the manuscript. We have now revised the Methods and Results accordingly to detail both parameters and significance. The revised text on lines 170-178 and 313-316 reads:

Methods:

“Lastly, we tested for correlated evolution among our three traits using an independent contrasts correlation model in an MCMC framework via a two-step process in BayesTraits v4.1.1 (Meade & Pagel 2024). First, we ran the ‘complex’ analysis, using a sample period of 1,000 with 1,010,000 iterati

---

## [Decision Letter · Decision Letter 1]

3 Jan 2025

Tracing the evolution of key traits in dorid nudibranchs

PONE-D-24-20562R1

Dear Dr. Layton,

We’re pleased to inform you that your manuscript has been judged scientifically suitable for publication and will be formally accepted for publication once it meets all outstanding technical requirements.

Kind regards,

Satheesh Sathianeson, Ph.D

Academic Editor

PLOS ONE

Additional Editor Comments (optional):

Reviewers' comments:

Reviewer's Responses to Questions

**Comments to the Author**

1. If the authors have adequately addressed your comments raised in a previous round of review and you feel that this manuscript is now acceptable for publication, you may indicate that here to bypass the “Comments to the Author” section, enter your conflict of interest statement in the “Confidential to Editor” section, and submit your "Accept" recommendation.

Reviewer #1: All comments have been addressed

2. Is the manuscript technically sound, and do the data support the conclusions?

Reviewer #1: Yes

3. Has the statistical analysis been performed appropriately and rigorously? 

Reviewer #1: Yes

4. Have the authors made all data underlying the findings in their manuscript fully available?

Reviewer #1: Yes

5. Is the manuscript presented in an intelligible fashion and written in standard English?

Reviewer #1: Yes

6. Review Comments to the Author

Reviewer #1: I appreciate the author's effort for the improvement of the manuscript. I have no further questions and recommend the manuscript for publication in PLOS One.

7. PLOS authors have the option to publish the peer review history of their article (what does this mean? ). If published, this will include your full peer review and any attached files.

**Do you want your identity to be public for this peer review?** For information about this choice, including consent withdrawal, please see our Privacy Policy .

Reviewer #1: No

---

## [Editor Report · Acceptance letter]

PONE-D-24-20562R1

PLOS ONE

Dear Dr. Layton,

I'm pleased to inform you that your manuscript has been deemed suitable for publication in PLOS ONE. Congratulations! Your manuscript is now being handed over to our production team.

Kind regards,

on behalf of

Dr. Satheesh Sathianeson

Academic Editor

PLOS ONE